

# Technical note: Comparing ozone production efficiency (OPE) of chemical mechanisms using chemical process analysis (CPA)

Katie Tuite[1], Alan M. Dunker[2], Greg Yarwood[1]

[1] Ramboll, 7250 Redwood Blvd., Suite 105, Novato, CA 94945, USA

[2] A. M. Dunker, LLC, 4041 Vendome Drive, Auburn Hills, MI 48326, USA

*Correspondence to*: Katie Tuite (ktuite@ramboll.com)

**Abstract.** Chemical mechanisms are critical to chemical transport models for air quality research and policy analysis. Several mechanisms are available and intercomparison, especially using metrics which reduce sensitivity to modeling scenario, is important for interpreting results and assessing uncertainties. Here, we investigate Ozone Production Efficiency

(OPE) as a comparison metric under conditions where nitrogen oxides ($NO_X$) are limited. OPE is the net number of ozone molecules produced per $NO_X$ molecule lost and can be computed in simulations using chemical process analysis (CPA). We compute OPE (OPE-CPA) for four chemical mechanisms (CB6r5, CB7r1, SAPRC07, RACM2) and find a similar response to varying anthropogenic emissions of volatile organic compounds (VOC) and $NO_X$. RACM2 consistently produces the largest OPE-CPA and differences between mechanisms are greatest at high VOC/$NO_X$ ratios. The high RACM2 OPE-CPA

is partially due to a slower OH + $NO_2$ rate and potentially to its treatment of $NO_X$ recycling. OPE-CPA is generally consistent with aircraft OPE measurements downwind of Houston but direct comparison is difficult due to uncertainties in deposition and VOC speciation. More recent OPE measurements are required to determine whether trends over time are consistent. OPE-CPA responds nonlinearly to $NO_X$ and increases at low $NO_X$ even as ozone production decreases. Using OPE to predict ozone response to $NO_X$ emissions reductions is therefore an oversimplification that will tend to overstate

ozone reductions. OPE-CPA is a viable metric to compare mechanisms, however, additional work would be helpful to define standardized conditions for comparisons.

## 1 Introduction

Three-dimensional Chemical Transport Models (CTMs) provide a representation of the atmospheric processes leading to the formation of secondary pollutants such as ozone ($O_3$) and particulate matter <2.5 μm ($PM_{2.5}$). Regulatory agencies use CTMs

as one of their tools to determine what anthropogenic emissions to control and by how much to achieve the U.S. National Ambient Air Quality Standards (NAAQS) for $O_3$ and $PM_{2.5}$. Key components of CTMs are the gas-phase chemical mechanisms that connect primary emissions to secondary pollutants. CTMs require efficient, condensed chemical mechanisms and multiple mechanisms are currently available for preparing U.S. emission control strategies, including the Carbon Bond version 6 revision 3 (CB6r3) (Emery et al., 2015); the Statewide Air Pollution Research Center 2007



(SAPRC07) (Carter, 2010a); and the Regional Atmospheric Chemistry Mechanism version 2 (RACM2) (Goliff et al., 2013). These mechanisms have been included in both the U. S. Environmental Protection Agency (EPA) Community Multiscale Air Quality Model (CMAQ) and the Comprehensive Air Quality Model with Extensions (CAMx). Current versions of CAMx include more recent versions of the Carbon Bond mechanism, CB6r5 (Yarwood et al., 2020) and CB7r1 (Yarwood et al., 2021).

Mechanism intercomparison is important to interpreting results and assessing uncertainties. Several recent studies compare $O_3$ formation when selected mechanisms are used in the same model with equivalent emissions for all mechanisms (Bates et al., 2021; Chen et al., 2024; Derwent, 2017; Derwent, 2020; Place et al., 2023; Shareef et al., 2022). Standardized metrics, such as the Maximum Incremental Reactivity (MIR) factor (Carter, 1994), are useful for mechanism comparisons since they reduce sensitivity to the modeling scenario. MIR is useful for comparing $O_3$ forming tendency of volatile organic compounds

(VOCs), $dO_3/dVOC$, under VOC-limited conditions. In recent years, $O_3$ formation in the U.S. has become limited on days exceeding the NAAQS by the availability of nitrogen oxides ($NO_X = NO + NO_2$) or trended toward this limitation, except in major urban centers (Blanchard and Hidy, 2018; Tao et al., 2022; Chen et al., 2023; Acdan et al., 2023). There is therefore a need for a comparison metric suitable for $NO_X$-limited conditions.

A key descriptor of $NO_X$-limited $O_3$ formation is the net Ozone Production Efficiency (OPE), which is the net number of $O_3$

molecules produced per $NO_X$ molecule lost (Kleinman et al., 2002). Here, net $O_3$ produced is the difference between $O_3$ produced by chemical reactions minus $O_3$ lost by reactions. In this study, we investigate OPE as a metric for comparing mechanisms under $NO_X$-limited conditions using a 2-box configuration of CAMx. Prior work using the Decoupled Direct Method (DDM) to calculate OPE (OPE-DDM) in 3D simulations encountered difficulties accounting for effects of deposition (Henneman et al., 2017). We investigated using OPE-DDM in this study but encountered non-intuitive results

such as computing zero OPE-DDM when $O_3$ was clearly increasing. Instead, we use chemical process analysis (CPA) to compute OPE and compare results among four chemical mechanisms – CB6r5, CB7r1, SAPRC07, and RACM2. Simulations were performed for three Texas cities during typical high ozone events during the 2019 ozone season. We also reviewed available measurements of OPE and conducted simulations to represent measurements years to compare measured and modeled OPE.

## 2 Methods

### 2.1 OPE measurement review

We reviewed the published literature from year 2000 forward to find OPE estimates from ambient measurements in the eastern U.S. for comparison to our modeled OPE results. We found OPE estimates from aircraft and surface measurements in various locations, the earliest measurements being in 2000 and the latest in 2023. We did not re-analyze any of the

measurements but used the OPE estimates obtained by the data collection teams.



Because our modeling is conducted for Texas cities, we focused on the aircraft measurements in transects of the Houston plume during the Texas Air Quality Study (TexAQS) 2000 (Ryerson et al., 2003; Daum et al., 2003; Daum et al., 2004; Zhou et al., 2014), TexAQS 2006 (Zhou et al., 2014; Neuman et al., 2009), and Deriving Information on Surface Conditions from Column and Vertically Resolved Observations Relevant to Air Quality (DISCOVER-AQ) in 2013 (Mazzuca et al., 2016). For a regional area in the southeast U.S. (not including Texas), Travis et al. (2016) estimated OPE from aircraft measurements during the Intercontinental Chemical Transport Experiment - North America (INTEX-NA) in 2004 and the Studies of Emissions and Atmospheric Composition, Clouds and Climate Coupling by Regional Surveys (SEAC4RS) campaign of 2013. Hembeck et al. (2019) give OPE estimates for the Baltimore area from aircraft flights during DISCOVER-AQ in 2011. Chace et al. (2025) estimated OPE from aircraft measurements in the urban plumes of New York City, Chicago, and Los Angeles in 2023 during the Atmospheric Emissions and Reactions Observed from Megacities to Marine Areas (AEROMMA) campaign. Data from surface sites over extended periods of one or more months have also been used to estimate OPE (Griffin et al., 2004; Blanchard and Hidy, 2018; Ninneman et al., 2017; Ninneman et al.,2019).

Net OPE can be estimated from atmospheric measurements by multiple methods. The most common method is to plot the $O_3$ or odd oxygen ($O_X$) concentration as a function of the $NO_Z$ concentration from collocated measurements. $NO_Z$ is usually determined as $NO_Z = NO_Y - NO_X$ with $NO_Y$ being the total reactive odd nitrogen. However, $NO_Z$ is sometimes determined by summing measurements of individual $NO_X$ oxidation products, e.g., $HNO_3$, PANs, and organic nitrates (ONs). If there is good correlation between the $O_3$ and $NO_Z$ concentrations, the slope of a linear regression of the data is an estimate of OPE, OPE-plot (Trainer et al., 1993). Comparisons of OPE-plot determined using $O_X$ and $O_3$ have shown only small differences (Neuman et al., 2009; Blanchard and Hidy, 2018). OPE-plot is termed an integrated estimate because it depends on the time-history of the air parcel prior to the measurements (Kleinman et al., 2002).

For aircraft transects across plumes, OPE can be estimated by integrating the $O_3$ and $NO_Z$ measurements across the plume and then calculating the ratio of the integrated $O_3$ and $NO_Z$ concentrations (Ryerson et al., 2003; Neuman et al., 2009). Another method used for plume transects is to determine the concentration differences of $O_3$ and $NO_Z$ between the plume center and edges and take the ratio of these differences as an estimate of OPE (Zaveri et al., 2003; Chace et al. 2025). These methods are usually applied only to well-defined plumes in relatively constant background concentrations and, as for the OPE-plot method, give integrated estimates of OPE over the history of the plume.

A quite different method uses predictions of a constrained steady-state (CSS) box or Lagrangian model (Kleinman et al. 2002; Daum et al., 2004; Zhou et al., 2014; Mazzucca et al., 2016). Atmospheric measurements of longer-lived species (e.g., $O_3$, CO, NO, $NO_2$, VOCs, HCHO, $H_2O_2$, $H_2O$), solar intensity, temperature, and pressure are used to fix the corresponding quantities in the CSS model. Once the radical species (e.g., OH, $HO_2$) achieve a steady state in the model, the formation and loss rates of $O_3$ and the formation rate of $NO_Z$ are obtained from the model reactions and concentrations, and OPE is estimated by the ratio of net $O_3$ formation to $NO_Z$ formation. This method relies upon the CSS model solution for short-lived species and consequently gives an instantaneous estimate of OPE at the time of the measurements as opposed to an integrated estimate over the history of the air parcel.





## 2.2 CAMx 2-box model


CAMx was configured as a 2-box model to compute OPE for three Texas locations, Houston-Galveston-Brazoria (HGB), Dallas-Fort Worth (DFW), and San Antonio (SAN). Each model scenario is 5 days and represents typical high-ozone summertime conditions for each location. We focus primarily on the HGB simulations to compare with available OPE measurements.

The CAMx 2-box model domain has 3 x 3 x 2 grid cells (in the x, y, and z dimensions) which is the smallest allowable domain in CAMx due to boundary condition and vertical transport requirements. All 9 grid cells in each layer have identical meteorologic input and a nominal 4 km grid size. The center grid cells, i.e., (2,2,1) and (2,2,2), form a 1D column of 2 boxes, with layer 1 representing the planetary boundary layer (PBL) and layer 2 representing a residual layer between the PBL and the CAMx top. Horizontal wind speeds in layer 1 are set to zero, preventing horizontal exchange between grid cells and

ensuring lateral boundary conditions have no influence. In layer 2, there is a constant horizontal wind speed to purge the layer with a 12-hour lifetime to limit the accumulation of pollutants over time.

Input data for the 2-box model scenarios were extracted from 3D CAMx simulations from the Texas Commission on Environmental Quality's (TCEQ) 2019 modeling platform (https://www.tceq.texas.gov/airquality/airmod/data/tx2019), which used meteorology from the Weather Research and Forecasting (WRF) model. The rectangular areas chosen to

represent the three locations are shown in Figure 1 and contain the central urban counties (Harris County for HGB, Dallas and Tarrant Counties for DFW, Bexar County for SAN) along with parts of adjacent counties. Data were averaged over the grid cells within these rectangular areas to provide the initial conditions, meteorology (temperature, humidity, PBL height; Figure S1), and emissions used in the 2-box model. The PBL height, as modeled by WRF, varies in time and is used to define the top of layer 1, whereas the top of layer 2 is constant in time at 3,000 m.





**Figure 1: The 4-km (red box) CAMx modeling domain used by TCEQ to model year 2019 in Texas. Data for the DFW, HGB, and SAN box model scenarios were extracted from the TCEQ modeling database for the rectangular regions surrounding these cities.**

Daily emissions of $NO_X$, anthropogenic VOC (AVOC), biogenic VOC (BVOC), and CO for the HGB, DFW, and SAN scenarios are provided in Table S1. Anthropogenic $NO_X$ ($ANO_X$) and AVOC emissions for years with available OPE measurements in the Houston area (Table 1) were used to interpolate box model results to measurement years. Since emission inventory methodologies have changed over the time period considered for this study, total $ANO_X$ and total AVOC emissions were used for the interpolation. VOC speciation is therefore constant over all years, consistent with 2019 emission speciation.

**Table 1: Anthropogenic $NO_X$ (ANOx) and anthropogenic VOC (AVOC) emissions from point and non-point sources for Harris County, TX for model year (2019) and years with available aircraft OPE measurements. Emissions data provided by the Texas Commission on Environmental Quality.**

| Year [a] | $ANO_X$ [b] | AVOC [b] | $AVOC/ANO_X$ |
|---|---|---|---|



| | Emissions (TPY) | Ratio to 2019 | Emissions (TPY) | Ratio to 2019 | (moleC/mole) |
|---|---|---|---|---|---|
| 2000 | 215,800 | 3.3 | 150,200 | 1.4 | 2.2 |
| 2006 | 153,630 | 2.3 | 134,000 | 1.3 | 2.8 |
| 2013 | 88,644 | 1.3 | 106,876 | 1.0 | 3.7 |
| 2019 | 66,340 | 1.0 | 104,960 | 1.0 | 5.0 |

[a] Point source emissions are year-specific; non-point source emissions are interpolated from data for 2002, 2005, 2008, 2011, 2014, and 2020.

[b] Emission inventory methodologies vary over time period shown.

## 2.3 Chemical mechanisms

Model simulations used gas-phase chemical mechanisms in CAMx version 7.2 (Emery et al., 2024), specifically the Carbon Bond mechanism versions 6 revision 5 and 7 revision 1 (CB6r5 and CB7r1; Yarwood et al., 2020; Yarwood et al., 2021), the toxics version of the Statewide Air Pollution Research Center 2007 mechanism (SAPRC07TC; Carter, 2010a; Carter, 2010b), and a version of the Regional Atmospheric Chemistry Mechanism version 2 provided by the mechanism developer in September 2021 (RACM2s21; Stockwell, 2021; Goliff et al., 2013). We coordinated with each mechanism's developer to ensure that they are implemented as intended. Coordination is particularly relevant to photolysis reactions and we use cross-section and quantum yield data provided by each mechanism developer, which we implemented into the Tropospheric Ultraviolet and Visible (TUV) radiative transfer model (NCAR, 2025) as a CAMx preprocessor (Ramboll, 2024).

We harmonized treatments of heterogeneous chemistry and iodine to focus on gas-phase reactions that relate $O_3$ and $NO_Z$. In CAMx, both CB6r5 and CB7r1 include a compact scheme (16 reactions) for $O_3$ destruction by oceanic iodine emissions (Emery et al., 2024) which we deactivated by zeroing photolysis frequencies of $I_2$ and HOI in the chemistry input file for these mechanisms. Ozone destruction by iodine can be several ppb/day for coastal locations such as Houston (Tuite et al., 2018). We deactivated the CAMx particle-phase and aqueous-phase chemistry in the chemistry input file for each mechanism. However, hydrolysis of $N_2O_5$ and ONs remained active for all mechanisms with consistent rate assumptions. With CAMx heterogeneous chemistry turned off, $N_2O_5$ hydrolyzes to $HNO_3$ at the bimolecular gas-phase rate (i.e., $N_2O_5$ + $H_2O$) measured by Wahner et al. (1998) and all ONs hydrolyze to $HNO_3$ with a lifetime of 12 hours derived from the experiments by Liu et al. (2012) and the ambient measurements of Rollins et al. (2013) although this ON hydrolysis lifetime may be a low estimate (Zhao et al., 2023).

The Supplement lists the reactions of each mechanism (Tables S4, S7, S10 and S13), their model species (Tables S5, S8, S11 and S14) and photolysis rates at representative conditions for several zenith angles (Tables S6, S9, S12 and S15). CB6r5 is the most compact (208 reactions and 80 species) followed by CB7r1 (214 reactions and 86 species), RACM2s21 (372 reactions and 117 species) and SAPRC07TC (567 reactions and 120 species). The major changes from CB6r5 to CB7r1 are a new scheme for isoprene (species ISOP) based on Wennberg et al. (2018), a new terpene scheme based on Schwantes et al. (2020) that separates α-pinene (APIN) from other terpenes (TERP), revised reactions of paraffinic alkoxy radicals (ROR)





that better differentiate how aldehyde and ketone formation depend on temperature and $O_2$ concentration, and less reactive
cresol (CRES) and aromatic ring-opening product (OPEN) to reduce reactivity of benzene (BENZ) and toluenes (TOL) in
better agreement with SAPRC07. Inorganic reaction rate constants were updated for CB6r5 and carried forward to CB7r1.
The mechanisms rely on different data sources for inorganic reaction rate constants. In general, SAPRC07 uses the NASA
(2006) recommendations, CB6r5 and CB7r1 follow IUPAC (2020), and RACM2s21 uses NASA (2019). We conducted a
sensitivity test where all mechanisms used the same OH + NO2 rate constant which reaffirmed the importance of this rate
constant to $O_3$ production.

## 2.4 Computing OPE with Chemical Process Analysis (OPE-CPA)

Model concentrations ($C_i$) of species $i$ change with time due to chemistry according to:

$$\frac{dC_i}{dt} = \sum_n s_{i,n} r_n ,$$

(1)

where the $r_n$ are rates (dimension concentration/time) of reactions involving species $i$ and the $s_{i,n}$ are stoichiometric
coefficients which must be multiplied by $-1$ for reactants. The $r_n$ depend on $C_i$ because they are computed as the product of
reactant concentrations and the reaction rate constant (or photolysis frequency) for each reaction. Time integration of the
coupled equations for $C_i$ and $r_n$ is performed by the CAMx chemistry solver, usually Hertel's enhancement of Euler's
method (Hertel et al., 1993). Process analysis captures the $r_n$ at each CAMx time step and accumulates them for output in
step with the model output for $C_i$. These integrated reaction rates can be subsequently analyzed to diagnose chemically
interesting quantities (termed process analysis) such as oxidant production rate or oxidant production sensitivity indicators
(Tonnesen and Dennis, 2000). However, these calculations are mechanism specific and can be complex to implement. CPA
internalizes these calculations within CAMx to directly output the chemically interesting quantities, which standardizes
methodology and is simpler to use.

We use CPA to compute the OPE for model simulations (OPE-CPA) as the ratio of net $O_3$ production $Pn(O_3)$ to net $NO_Z$
production $Pn(NO_Z)$ from start time $t_1$ to end time $t_2$:

$$\textbf{OPE-CPA} = \left[\frac{Pn(O_3)}{Pn(NO_z)}\right]_{t1}^{t2} ,$$

(2)

where net species production rate (Pn) signifies the net effect of chemical production combined with loss, and is computed
within CAMx from integrated reaction rates ($irr_n$) as:

$$Pn_i = \sum_n s_{i,n} irr_n ,$$

(3)

We take $t_1$ as the first hour after sunrise with positive $Pn(O_3)$ and $t_2$ as hour 15 (15:00 to 16:00 local time) which is consistent
with flight times that measured OPE near Houston (discussed above) and encompasses hours with maximum $O_3$ production
in our model simulations, as shown below. We used the same $t_2$ for other cities for comparability. Details of calculating
$Pn(O_3)$ and $Pn(NO_Z)$ for each mechanism are provided in the Supplement.





The *irr* values are local to each CAMx grid cell meaning that they are not directly influenced by model transport (advection and diffusion) or deposition processes. Transport and deposition indirectly affect chemistry by changing species concentrations and therefore can also indirectly affect *irr* values. Here, CAMx is configured as a 2-box model with the top of layer 1 following the PBL height provided by WRF. The aircraft flights that measured OPE near Houston were conducted within the PBL and therefore comparable to OPE-CPA for our CAMx layer 1. The change in PBL depth between $t_1$ and $t_2$ is

accounted for when computing OPE-CPA by a weighting factor ($PBL_t/PBL_{max}$):

$$[\text{OPE-CPA}]_{t1}^{t2} = \sum_{t1}^{t2}\left(\frac{PBL_t}{PBL_{max}}\right)Pn(O_3) / \sum_{t1}^{t2}\left(\frac{PBL_t}{PBL_{max}}\right)Pn(NO_z) , \tag{4}$$

where $PBL_{max}$ is the largest PBL depth between $t_1$ and $t_2$. This weighting considers that a deeper PBL contains more air mass and therefore contributes proportionately more to net species production within the time period analyzed. We applied Eq. 4 as a post-processing step using hourly-averaged $P_n$ obtained from CPA and the PBL depth from WRF.

**3 Results and discussion**

**3.1 OPE measurements**

We focus on the OPE-plot estimates for two reasons. First, more estimates are available from the OPE-plot method than the plume integration or plume center-edge methods. Second, CSS models require a chemical mechanism, and thus the OPE estimates depend on the mechanism employed. This adds uncertainty to the OPE estimates, which is difficult to assess

because different mechanisms have been used in different modeling studies and all the mechanisms used are older than those being compared in this work.

Table 2 gives OPE-plot estimates determined from aircraft measurements in Houston, Texas in 2000, 2006, and 2013. The daily time period of the measurements differs from study to study but generally contains the early to middle-afternoon hours with 15:00 or 16:00 being the endpoint of most periods. The OPE estimates for the industrial plumes in Houston in 2000

(~11) are about twice as large as those for the urban plumes (~5). This is attributed to large emissions of highly reactive VOCs (HRVOCs) from the petrochemical industry increasing OPE by forming $O_3$ efficiently in downwind plumes (Ryerson et al., 2003; Daum et al., 2003). The coalesced industrial and urban plumes in 2000 had OPEs similar to those of the urban plumes that year. The OPEs for the coalesced industrial and urban plumes were essentially the same in 2006 as in 2000, which may be due to offsetting effects of emissions reductions. There were large reductions in Houston's $NO_X$ emissions

from 2000 to 2006 and in HRVOC emissions from petrochemical facilities (Zhou et al., 2014) due to the HRVOC Emissions Cap and Trade (HECT) Program (TCEQ, 2025). The reduction in HRVOC emissions should reduce OPE, but a reduction in emissions and atmospheric concentrations of $NO_X$ generally increases OPE (Kleinman et al., 2002; Mazzuca et al., 2016; Henneman et al., 2017). The OPE for the industrial plume in 2006 is about 20% smaller than the OPEs for the industrial plumes in 2000, consistent with the reduced HRVOC emissions. The Houston coalesced industrial and urban plumes in 2013





had an OPE of 8, which is 35% - 60% larger than the estimates for 2006. The increase in 2013 might be due to the

continuing $NO_X$ emission reductions in Houston.

**Table 2: Estimates of net OPE from aircraft measurements in Houston, TX.**

| Measurement program [a] | Plume type | Altitude (m) | Date | Time (CST) [b] | OPE-plot [c] | Reference |
|---|---|---|---|---|---|---|
| TexAQS 2000 | urban | 400 - 700 | 28 Aug. 2000 | 1400 - 1500 | $5.4 \pm 0.2$[d] | Ryerson et al. (2003) |
| | coalesced industrial | 400 - 700 | 27-28 Aug. 2000 | 1400 - 1500 | 11-12[e] | |
| TexAQS 2000 | urban | 500 - 750 | 29 Aug. 2000 | 1300 - 1600 | 5.1[f] | Daum et al. (2003) |
| | industrial | 500 -750 | 29 Aug. 2000 | 1300 - 1600 | 10.9[f] | |
| TexAQS 2000 | coalesced industrial and urban | 500 - 750 | 19 Aug. -6 Sept. 2000 | 1300 - 1600 | $6.4 - 11$[e,f] | Daum et al. (2004) |
| TexAQS 2000 | coalesced industrial and urban | 400 -700 | 20 Aug - 10 Sept. 2000 | 1200 - 1700 | $5.3 \pm 1.1$[g] | Zhou et al. (2014) |
| TexAQS 2006 | coalesced industrial and urban | 400 -700 | 13 Sept. - 10 Oct. 2006 | 1300 - 1800 | $4.9 \pm 1.4$[g] | |
| TexAQS 2006 | coalesced industrial and urban | ~500 | 25 Sept. 2006 | 1600 - 1715 | $5.2 - 6.7$[e] | Neuman et al. (2009) |
| | coalesced industrial and urban | ~500 | 20, 25, 26 Sept., 5 Oct. 2006 | 1300 -1800 | $5.9 \pm 1.2$[g] | |
| | industrial | ~500 | 6 Oct. 2006 | 1300 - 1500 | 9 | |
| DISCOVER-AQ | coalesced industrial and urban | ~250 - 1000 | 4 -29 Sept. 2013 | 0900 - 1500 | 8.0[f] | Mazzuca et al. (2016) |

[a] TexAQS = Texas Air Quality Study; DISCOVER-AQ = Deriving Information on Surface Conditions from Column and Vertically Resolved Observations Relevant to Air Quality

[b] Approximate time period of the measurements based on information in the references

[c] $O_3$ used to determine OPE unless otherwise indicated

[d] Uncertainty from the linear fit of $O_3$ to $NO_Z$ data

[e] Range for multiple transects/plumes

[f] $O_X = O_3 + NO_2$ used instead of $O_3$

[g] Average over multiple transects

The OPE-plot estimates in Table S2 from the regional INTEX-NA and SEAC4RS flights over the southeast U.S. (14 and 17 respectively) are significantly larger than all the estimates for Houston. This difference likely results from the lower $NO_Z$ concentrations measured in the regional flights than in the Houston flights. The smaller $NO_Z$ concentrations could be due to

greater dilution of $NO_X$ emissions by background or rural air or greater deposition of $NO_Z$, which would increase the OPE estimates. The OPE estimates from DISCOVER-AQ for the Baltimore urban area in 2011 ($8.4 \pm 4.1$) and the estimates for for New York City ($9 \pm 4$) and Chicago ($6 \pm 3$) in 2023 from AEROMMA are similar to those for the Houston coalesced industrial and urban plume in 2013 (8.0). However, the $NO_X$ concentrations and also likely the VOC concentrations vary among these urban plumes, and consequently the similar OPE values do not imply that the chemistry is similar in the

plumes. Again, increased $NO_X$ and increased VOC emissions can have opposing effects on OPE that cancel.

The OPE-plot estimates for many surface sites in Table S3 are also larger than the estimates for Houston in Table 2. This is expected for rural sites because the $NO_X$ or $NO_Z$ concentration is generally smaller than in Houston, leading to larger OPE





estimates. At Whiteface Mt., for example, the median $NO_X$ concentration was only 0.2 ppb. Similarly, the urban and suburban SEARCH sites have smaller $NO_Z$ concentrations than the Houston measurements. The OPE estimates for Durham and Flushing are comparable to that for Houston in 2013, consistent with similar $NO_Z$ concentrations at these locations.

The OPE values in Table 2, S2 and S3 have additional uncertainties that are not reflected in the shown uncertainties. The most important is the amount of $NO_Z$ species, particularly $HNO_3$, deposited prior to the measurements. If this deposition is significant, OPE-plot is an upper limit to the OPE determined by the chemistry alone. Also, OPE-plot may not represent $O_3$ formation in a single air parcel because the measurements may sample different air parcels containing emissions from different sources or sample an air parcel that is a mixture of multiple air parcels with different photochemical ages. These complications can introduce significant scatter and nonlinearity into the $O_3$ vs. $NO_Z$ relationship that alters the linear regression of the data. OPE-plot from surface data is more strongly influenced by these uncertainties than OPE-plot from aircraft flights because analyses of surface data usually combine data from many days and different air parcels whereas flight transects focus on a specific air parcel over a short time period, and surface measurements are more likely to be affected by $HNO_3$ deposition.

## 3.2 Model base cases and $O_3$ response surfaces

Four chemical mechanisms were evaluated using results from the CAMx 2-box model scenarios. We focus on the Houston (HGB) scenario due to the availability of OPE measurements at this location and investigate mechanism differences in $O_3$, $NO_Z$ ($HNO_3$ + ONs + PANs), and $NO_Y$ ($NO_Z$ + $NO_X$), which are most relevant to OPE. Results from the other model locations, DFW and SAN, are provided in the Supplement for comparison.

Time series of hourly average $O_3$ and $NO_Z$ concentrations over the 5-day model period are shown in Figure 2. The diurnal trends for both species are similar between mechanisms. $O_3$ increases throughout the day and peaks in the late afternoon, whereas $NO_Z$ increases throughout the day and overnight, peaks in the early morning, and decreases sharply as the PBL grows. The buildup in $O_3$ concentration over the first two days results from carryover of $O_3$ via the residual layer. The accumulated $O_3$ in the residual layer (layer 2 of the model) is ventilated to background air and/or entrained into the PBL (layer 1) and the concentration stabilizes after 2 days. CO (Figure S2) shows a similar trend as $O_3$. Day 1 is considered model spin-up and we focus on model days 2 through 5 so that initial conditions have minimal importance and emissions have maximum importance in the simulations.





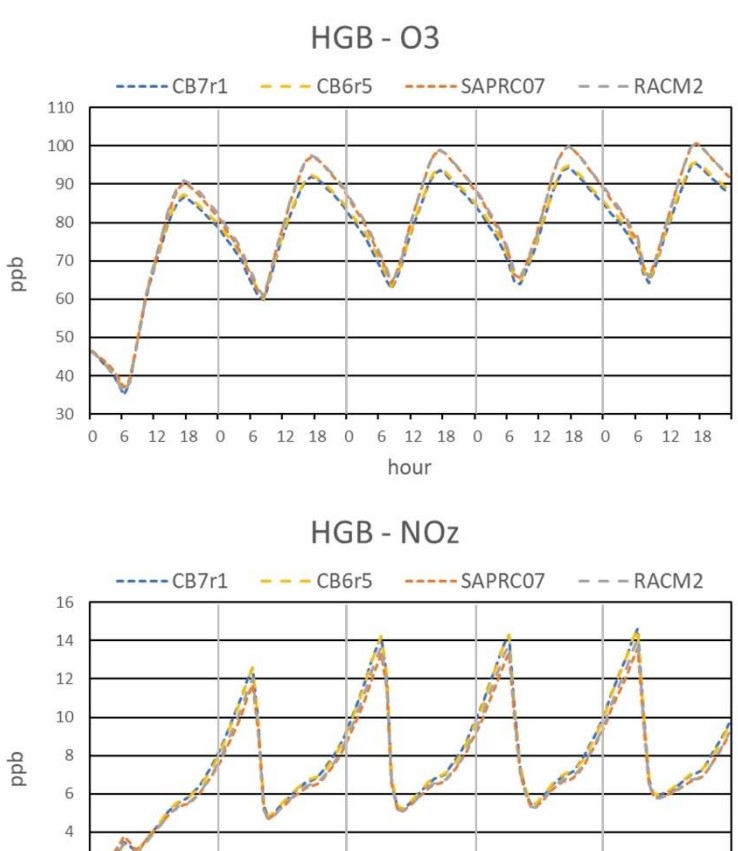

**Figure 2: Time series of O₃ and NOz (HNO₃+ONs+PANs) simulated by four chemical mechanisms for the HGB box model scenario, shown in local time from Sept 3-7, 2019.**

RACM2 and SAPRC07 predict higher daytime $O_3$ compared to CB6r5 and CB7r1, by about 5 ppb at the time of peak $O_3$ in late afternoon. The DFW model scenario (Figure S3) shows similar results but $O_3$ concentrations in the SAN scenario (Figure S4) agree closely between mechanisms. $NO_X$ concentrations are higher at HGB and DFW, indicating that RACM2 and SAPRC07 may produce $O_3$ more efficiently in high $NO_X$ environments. We investigated the importance of the OH + $NO_2$ = $HNO_3$ reaction to $O_3$ differences by performing a sensitivity test where all mechanisms use the same rate constant. The RACM2 rate was changed to the NASA (2006) recommendation used in CB6r5, CB7r1, and SAPRC07. Note that the NASA recommended rate for the OH + $NO_2$ = $HNO_3$ reaction remained unchanged from 2006 to 2019. Results of this test for HGB are shown in Figure S5. RACM2 OH and $O_3$ decrease and become more like CB6r5 and CB7r1, indicating that the difference in rate is a significant contributor to the higher RACM2 OH and $O_3$. There is also a 6% difference between current NASA (2006; 2019) and IUPAC (2020) recommendations which is a meaningful uncertainty that should be resolved (Amedro et al., 2020).



All mechanisms produce similar levels of daytime $NO_Z$, with slightly higher values from CB6r5 and CB7r1. $NO_Y$ concentrations are also similar between mechanisms but there are some differences in composition. $NO_Y$ composition at 15:00 on each model day is shown in Figure 3 for each mechanism. $HNO_3$ is the dominant $NO_Y$ species, highlighting the importance of the $OH + NO_2$ reaction in $NO_Z$ net production, $Pn(NO_Z)$, for all mechanisms. OH and $NO_2$ concentrations (Figure S2) are highest from RACM2, resulting in slightly higher $HNO_3$. While $NO_2$ is similar among the other three mechanisms, SAPRC07 has lower OH and consequently lower $HNO_3$. As noted above, the rate constants for the $OH + NO_2$ reaction also vary between mechanisms, contributing to differences in $HNO_3$. Concentrations of total ONs and peroxyacyl nitrates (PANs) vary between mechanisms. ON is lowest from RACM2 and this nitrogen is shifted to other $NO_Y$ species resulting in higher NO, $NO_2$, and PANs. Daytime ONs from RACM2 also remain relatively constant whereas the other mechanisms show increasing concentration throughout the day, consistent with RACM2 recycling more ONs to $NO_X$ than other mechanisms. Daytime PANs are highest from SAPRC07 and lowest from CB6r5 and CB7r1. Daytime NO is highest from RACM2 due to higher daytime $NO_2$ and rapid interconversion via the Leighton cycle (Leighton, 1961). Higher NO contributes to higher OH and lower $HO_2$ for RACM2 due to the $NO + HO_2 = NO_2 + OH$ reaction. The higher OH for RACM2, which is consistent across all three locations, will influence how many pollutants are removed in RACM2 compared to the other mechanisms.

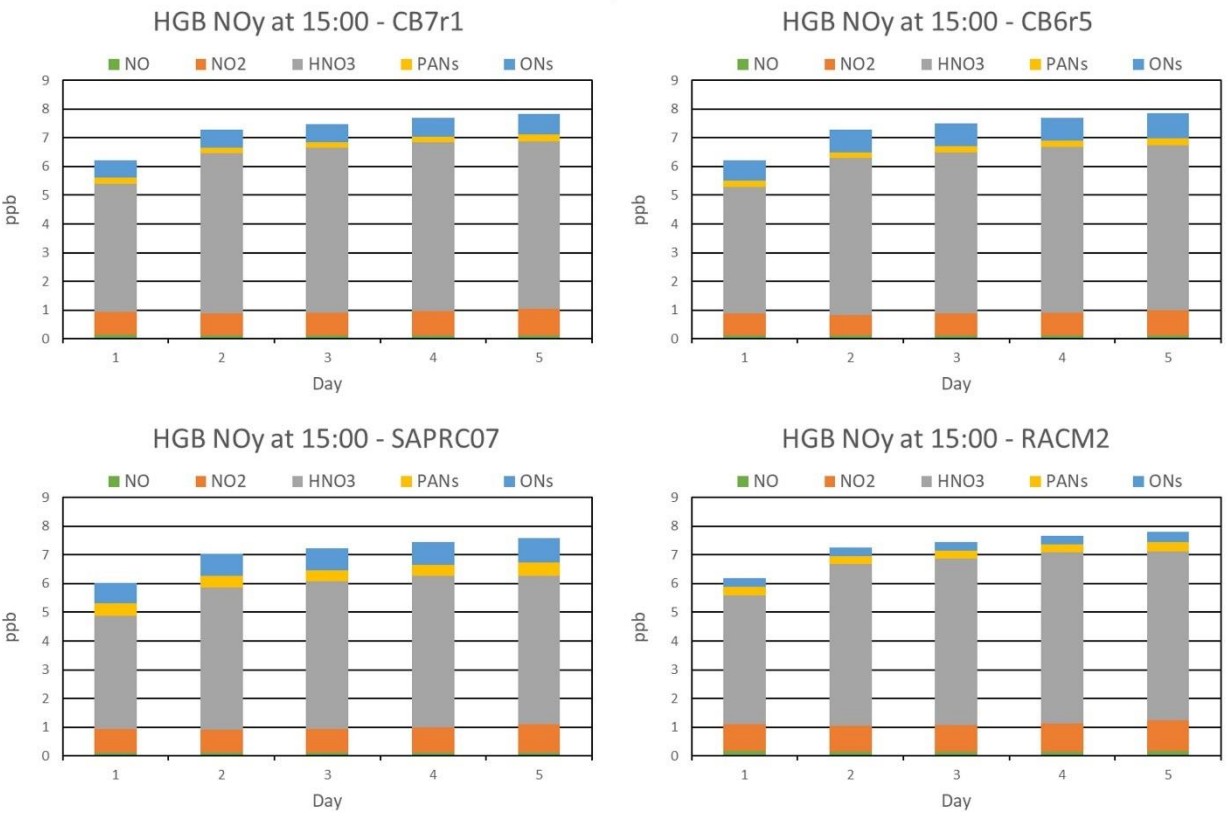



**Figure 3: NO$_Y$ composition simulated by four chemical mechanisms for the HGB box model scenario, shown at 15:00 (local time)**
**for each modeled day.**

In addition to base case simulations, we investigated how varying ANO$_X$ and AVOC emissions impact O$_3$ and OPE-CPA by performing a matrix of 196 box model simulations. Scale factors of 0.1 to 5.0 were applied to the base emissions and Figure 4 shows resulting O$_3$ response surface plots (for O$_3$ at 15:00 local time) for the four mechanisms. The base cases are at NO$_X$ and VOC scale factors of (1,1) and are in a NO$_X$ limited regime since O$_3$ reduces more rapidly with NO$_X$ reductions than

with VOC. As in the base case, RACM2 and SAPRC07 show higher O3 across all emission scales, but all mechanisms show a similar response shape. In particular, the location of the 'ridgeline', which separates NO$_X$ limited from VOC-limited conditions, is similar between mechanisms. Scale factors below 1.0 are relevant to near-term air quality planning purposes since existing strategies are expected to reduce emissions, particularly of NO$_X$. For all mechanisms, O$_3$ formation in this range is in a NO$_X$ limited regime indicating that all mechanisms find NO$_X$ emission reductions will be more effective than

VOC reductions for HGB as well as DFW and SAN (see the Supplement).



**Figure 4: O$_3$ response surface plots at varying anthropogenic VOC and anthropogenic NO$_X$ emissions for four chemical mechanisms, with the star indicating the base case. O$_3$ at 15:00 local time for day 3 (Sept 5, 2019) of the HGB box model scenario is shown. Other modeled days show similar O$_3$ responses.**

Overall, our results show relatively good agreement among the mechanisms consistent with Derwent (2017; 2020) and Shareef et al. (2022) but different from the lower O$_3$ formation found by Chen et al., (2024) for CB6r2. The reason for the low O$_3$ formation with CB6r2 in the Chen et al. (2024) work is unclear.

### 3.2 OPE-CPA comparison

OPE-CPA was computed from the matrix simulations using the method described in the Sect. 2.4. Transects of OPE-CPA at
varying NO$_X$ and VOC scaling factors are presented in Figure 5 and Figure 6, respectively. In general, OPE-CPA for each of the mechanisms responds similarly to varying emissions, increasing as VOC increases and decreasing as NO$_X$ increases. At high VOC/NO$_X$ ratios, OPE-CPA can peak or plateau but this behavior is not consistent from day to day or among mechanisms. This aligns with inconsistencies in measurements (Ninneman et al., 2017; Blanchard and Hidy, 2018) and prior modeling studies (Kleinman et al., 2002; Mazzuca et al., 2016; Henneman et al., 2017), some of which observed a peak or
plateau and others did not. Regardless of OPE behavior, however, O$_3$ in our simulations continues to decrease as NO$_X$ decreases (Figure 4).





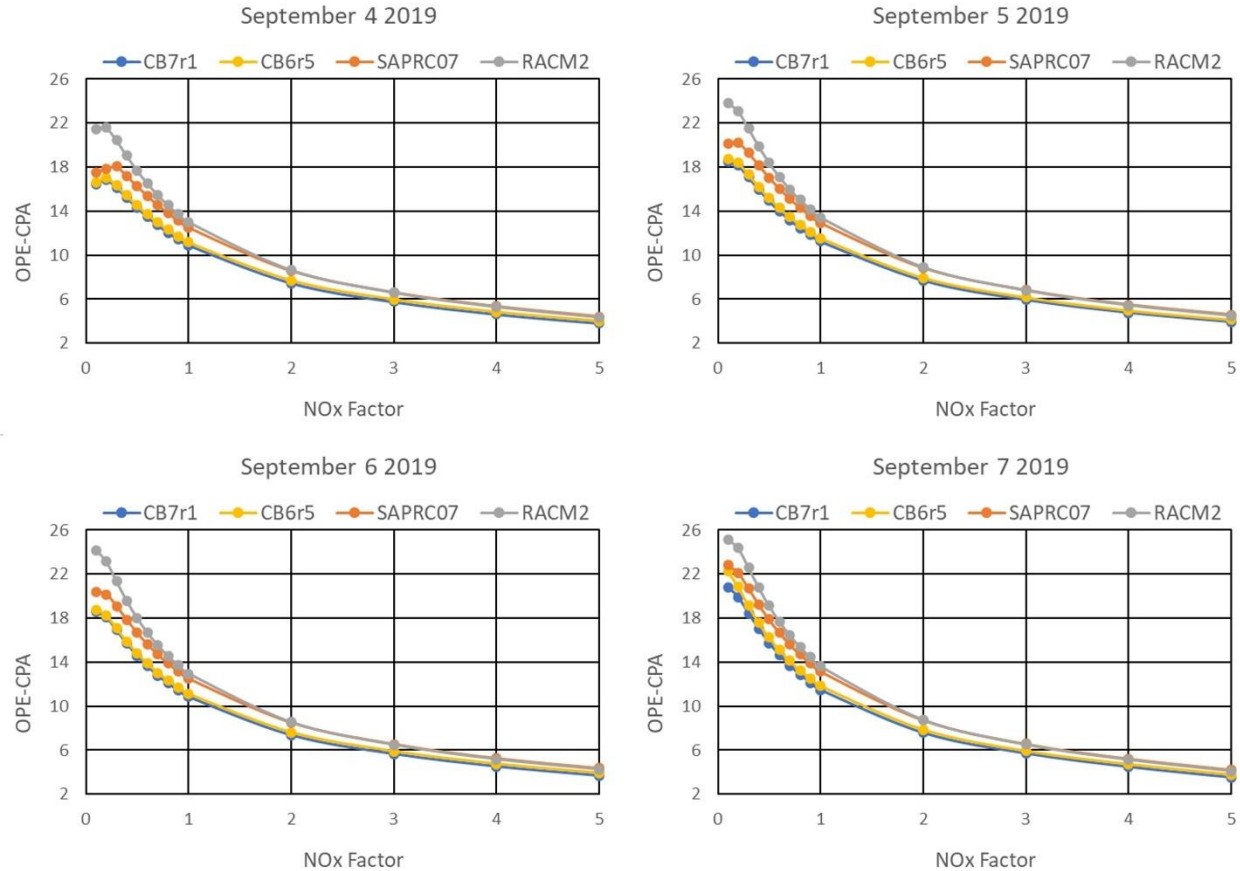

**Figure 5: OPE-CPA calculated with $t_2$=15:00 (local time) at varying anthropogenic NO$_X$ emission scaling factors and base VOC emissions, simulated by four chemical mechanisms for the HGB box model scenario.**




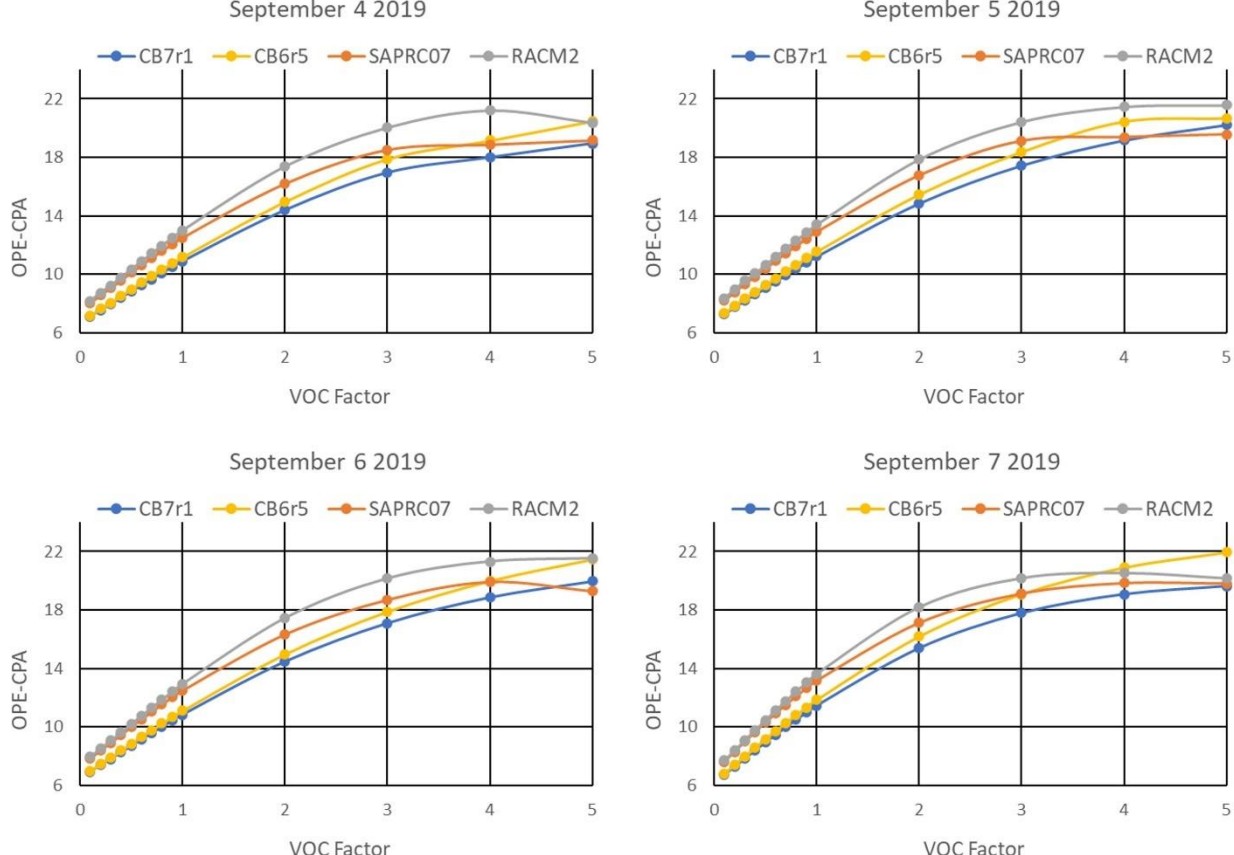


**Figure 6: OPE-CPA calculated with $t_2$=15:00 (local time) at varying anthropogenic VOC emission scaling factors and base NO$_X$ emissions, simulated by four chemical mechanisms for the HGB box model scenario.**

RACM2 consistently has the highest OPE-CPA across all NO$_X$ and VOC scales and CB7r1 has the lowest. As discussed in the previous section, HNO$_3$ is the largest component of NO$_Z$ and NO$_Y$ (Figure 3), so OH + NO$_2$ dominates Pn(NO$_Z$). The

slower OH + NO$_2$ rate in RACM2 contributes to the higher OPE-CPA, as is evident from our sensitivity test which normalized the rate among all four mechanisms. When the RACM2 rate was adjusted to match the other mechanisms, OPE-CPA decreased by about 7% in the HGB base case simulations, putting it between values for SAPRC07 and CB7r1. OPE-CPA also decreased across all NO$_X$ scaling factors (Figure S14) but is still higher than the other mechanisms at low NO$_X$ on all model days.

Another factor that may play a role in the OPE-CPA differences is NO$_X$ recycling. Differences between mechanisms are largest at high VOC/NO$_X$ ratios (NO$_X$ factor <1 in Figure 5 and VOC factor >1 in Figure 6) where O$_3$ formation is strongly limited by NO availability and NO$_X$ recycling becomes more important. The mechanism differences at NO$_X$ factor <1 are particularly important to note since this may be relevant to air quality planning. RACM2 allows all ONs to recycle nitrogen



to $NO_X$ but the other mechanisms include ON species (XN in SAPRC07 and NTR2 in CB6r5 and CB7r1) which do not
recycle. This difference in $NO_X$ recycling may contribute to higher OPE-CPA for RACM2 under $NO_X$-limited conditions.

Table 3 provides a comparison of OPE-CPA to OPE-plot calculated from measurements near Houston. Model results were
interpolated to measurement years using the emissions trends shown in Table 1. We assume model emissions represent a
combination of general urban and industrial emissions so that model comparisons to urban + industrial measurements are
most appropriate. OPE-CPA is similar to the urban + industrial measurements in 2000 but greater than those in 2006 and
2013. One uncertainty in the comparison relates to how VOC emissions have changed from 2000 to 2019. In particular,
emissions of highly reactive VOCs (HRVOCs) declined by 40% from 2000 to 2006 due to targeted reductions from
industrial sources (Zhou et al., 2014) and likely have remained lower through 2019. However, our model VOC speciation is
constant over all years and representative of 2019, so changes in HRVOCs are not captured. Since higher HRVOC
concentrations are expected to increase OPE, our OPE-CPA may be an underestimate for the measurement years,
particularly for 2000. The higher OPE in industrial plumes in Table 3 are likely due to increased levels of HRVOCs and/or
higher VOC/$NO_X$ ratios. 3D modeling is better suited than box modeling to further investigate how VOC/$NO_X$ ratios vary
between plumes.

**Table 3: Comparison of modeled to measured OPE for Houston.**

| Mechanism | Year | OPE-CPA[a] | OPE-plot for plume type | | |
|---|---|---|---|---|---|
| | | | Urban | Industrial | Urban + industrial |
| CB7r1 | 2000 | 6.2 | 5.4 ± 0.2[b] 5.1[c] | 11-12[b] 10.9[c] | 6.4 – 11[d] 5.3 ± 1.1[e] |
| CB6r5 | | 6.5 | | | |
| RACM2 | | 7.1 | | | |
| S07 | | 7.1 | | | |
| CB7r1 | 2006 | 7.7 | --- | 9[e] | 4.9 ± 1.4[e] 5.2 - 6.7[f] 5.9 ± 1.2[f] |
| CB6r5 | | 8.0 | | | |
| RACM2 | | 9.0 | | | |
| SAPRC07 | | 8.9 | | | |
| CB7r1 | 2013 | 10.0 | --- | --- | 8.0[g] |
| CB6r5 | | 10.3 | | | |
| RACM2 | | 11.9 | | | |
| SAPRC07 | | 11.5 | | | |

[a] Calculated with $t_2$=15:00 (local time) and averaged over model days 2-5 (Sept 4-7, 2019); Harris County emission trends are used to interpolate CPA-OPE
from model year (2019) to measurement years (2000, 2006, 2013).

[b] Ryerson et al. (2003)

[c] Daum et al. (2003)

[d] Daum et al. (2004)

[e] Zhou et al. (2014)

[f] Neuman et al. (2009)

[g] Mazzuca et al. (2016)




Another important difference between OPE-CPA and OPE-plot is the influence of $NO_Z$ deposition. Our OPE-CPA is only indirectly affected by deposition (see Sect. 2.4) but OPE-plot is directly influenced by deposition, although with less impact for these aircraft measurements than for surface measurements as discussed above. Because of this, comparison between

OPE-CPA and OPE-plot is difficult. Still, considering the range in OPE-plot, OPE-CPA values are reasonable and the differences may not be significant given the uncertainties. Insufficient measured OPE data over time also make it difficult to determine whether trends are consistent.

The impact of $NO_X$ reductions on OPE-CPA and $O_3$ are shown in Table 4. In model runs where $ANO_X$ emissions are reduced by 50%, OPE-CPA increases by 32-38% depending on the mechanism, and $O_3$ decreases by 14-17%. The increase

in OPE-CPA is noticeably larger for RACM2, consistent with the transects shown in Figure 5. The higher OPE-CPA for RACM2 for both the base and 50% $ANO_X$ cases corresponds to higher $O_3$. SAPRC07 has the smallest relative change in OPE-CPA but the largest change in $O_3$. The fact that each mechanism shows a similar OPE dependence on $NO_X$ emissions and predicts similar reductions in $O_3$ is reassuring from a regulatory modeling perspective.

**Table 4: Comparison of simulated CPA-OPE and $O_3$ (at 15:00 local time) from four chemical mechanisms for HGB between base**
**emission scenario and 50% anthropogenic $NO_X$ ($ANO_X$) emission scenario.**

| Mechanism | OPE-CPA [a] | | | $O_3$ (ppb) [a] | | |
|---|---|---|---|---|---|---|
| | Base Run | 50% $ANO_X$ | % Difference | Base Run | 50% $ANO_X$ | % Difference |
| CB7r1 | 11.1 | 14.9 | 34.0% | 87.0 | 74.1 | -14.8% |
| CB6r5 | 11.4 | 15.2 | 33.0% | 87.4 | 74.4 | -14.9% |
| RACM2 | 13.2 | 18.3 | 38.2% | 91.2 | 77.0 | -15.6% |
| SAPRC07 | 12.8 | 17.0 | 32.8% | 91.0 | 76.1 | -16.4% |

[a] Averaged over model days 2-5.

OPE-CPA increases as $NO_X$ decreases but, counterintuitively, $O_3$ still decreases (see Figure 4 and Figure 5). Also, the percent increase in OPE-CPA for a 50% reduction in $ANO_X$ is about twice as large as the percent $O_3$ decrease (Table 4). $Pn(O3)$ is impacted by factors other than OPE (e.g., VOC oxidation rate) which also depend on $NO_X$. The difference in the

relative changes of OPE and $O_3$ indicate that using OPE to predict $O_3$ response to $NO_X$ emissions would be an over-simplification that will tend to over-state $O_3$ reductions. This may be especially true at low $NO_X$ where the mechanisms have the largest variation in OPE-CPA.

As discussed in the Sect. 2, OPE-plot is derived from a linear relationship between $O_3$ and $NO_Z$, which depends on $NO_X$. OPE-CPA, on the other hand, varies nonlinearly with $NO_X$, as seen in Figure 5. It is unclear why a linear relationship of $O_3$

and $NO_Z$ is observed in measurements despite a nonlinear relationship between $Pn(O_3)$ and $Pn(NO_Z)$ (Kleinman et al., 2002). Additional studies which focus on the influence of plume dilution, composition of background air, and variations of VOC/$NO_X$ within plumes are needed to explain why OPE-plot and OPE-CPA behave differently. For example, by conducting 3D simulations with finely resolved grids and emission data, sub-hourly OPE-plot and OPE-CPA computed along pseudo aircraft trajectories could be compared.





## 4 Conclusions

### 4.1 Summary of results and uncertainties

We performed CAMx 2-box model simulations with four widely used chemical mechanisms (CB6r5, CB7r1, RACM2, and SAPRC07) and computed OPE using chemical process analysis (OPE-CPA). In general, we found relatively good agreement between the mechanisms for $O_3$, $NO_Z$, $NO_Y$, and OPE-CPA at all three Texas locations. There was better $O_3$ agreement at

SAN compared to HGB and DFW, indicating that mechanism differences in $O_3$ production are greater in high $NO_X$ environments. Higher values of $O_3$, OH, and OPE-CPA from RACM2 are partially due to a slower $OH + NO_2$ rate constant compared to the other mechanisms. $OH + NO_2$ is important to $O_3$ chemistry and dominates $Pn(NO_Z)$ so it plays a key role in OPE. Sensitivity tests for HGB showed better agreement when a consistent rate was applied for all mechanisms. Different rate constant recommendations from IUPAC and NASA can contribute to overall mechanism uncertainty, particularly via the

important $OH + NO_2$ reaction, demonstrating that new rate constant measurements are valuable (e.g., Rolletter et al., 2025; Amedro et al., 2020) together with updated rate constant recommendations. It is noteworthy that uncertainties in extensively studied inorganic reactions continue to be among the larger known uncertainties in chemical mechanisms.

We investigated how varying $NO_X$ and VOC emissions impact $O_3$ and OPE-CPA and found similar responses among all mechanisms. $O_3$ response surfaces show that the base emissions scenarios are in a $NO_X$ limited regime for all three locations.

OPE-CPA is inversely related to $NO_X$ and differences between mechanisms are greatest at high $VOC/NO_X$ ratios. In addition to the $OH + NO_2$ rate contributing to higher RACM2 OPE-CPA, the treatment of $NO_X$ recycling, which varies between mechanisms, may also play a role. The increase in OPE-CPA at low $NO_X$ counterintuitively occurs even as $O_3$ production decreases, and the relative changes are notably different, e.g., the OPE-CPA percent increase is 2 times larger than the $O_3$ percent decrease at 50% $ANO_X$. This highlights the difficulty of using OPE to predict $O_3$ response to $NO_X$.

The fact that all mechanisms show a similar dependence of OPE and $O_3$ to $NO_X$ emissions, however, does indicate that OPE-CPA can be used to compare mechanisms. Unlike Maximum Incremental Reactivity (MIR) factors though, which can be used to characterize $O_3$ formation under specific VOC-limited conditions, there is no obvious emission condition to compare OPE-CPA. We recommend further studies to investigate whether a suitable condition (perhaps, for example, 50% of peak OPE) exists to better utilize OPE-CPA as a comparison factor. This is especially important due to the limitations of MIR for

$NO_X$-limited conditions, which are common in many regions in the U.S. and relevant for air quality planning purposes.

OPE-CPA from the HGB simulation was also compared to available measurements (OPE-plot) in the Houston area. We focused on aircraft OPE measurements since surface measurements are subject to large uncertainties from deposition. Comparison to DFW and SAN simulations were not possible due to lack of measurements. While OPE-CPA was in relatively good agreement with OPE-plot, there are aspects which make comparison difficult, including uncertain VOC

speciation and impacts of dilution and $NO_Z$ deposition. The limited number of OPE measurements also restrict our ability to make conclusions about OPE trends over time. Additional aircraft based OPE measurements downwind of previously studied locations would be useful to test mechanism response to emission reductions, and speciated VOC measurements



would help characterize the reactivity of emissions. Clear reporting of the time of day for OPE measurements would also reduce uncertainty in comparisons between OPE-CPA and OPE-plot.

### 4.2 Potential future work

Applying OPE-CPA in 3D simulations is feasible and complementary with other methods used to probe 3D model simulations, such as sensitivity analysis. CPA can reveal spatial variations in chemical conditions between grid cells that are less apparent using sensitivity analysis due to the influence of transport. 3D simulations of urban plumes using a fine horizontal grid resolution could investigate why measured OPE is often stable within a plume even when subject to varying $NO_X$ emissions. Comparison of OPE-CPA and OPE-plot along pseudo aircraft transects in the same simulated plume would help us better understand if the two provide similar estimates of OPE. In contrast to a box model, 3D simulations may also place different emphasis on pollution carryover versus same day chemistry and the importance of PANs and ONs versus $HNO_3$. On a regional scale, the difference in ON and PAN chemistry between mechanisms may lead to differences in $O_3$ production if increased ON and PAN levels allow $NO_Y$ to be transported away from local emission sources and returned as $NO_X$ via photochemical reactions.

### Code and data availability

Data are provided in the manuscript and the Supplement. The CAMx code, open-source user license, release notes, and user guide documentation are publicly available at www.camx.com.

### Author contribution

Conceptualization, G.Y. and A.M.D.; methodology, G.Y. and A.M.D.; software, G.Y. and K.T.; validation, G.Y. and A.M.D.; formal analysis, K.T. and G.Y.; investigation, K.T. and G.Y.; resources, G.Y.; data curation, K.T.; writing – original draft preparation, K.T. and A.M.D.; writing – review and editing, G.Y. and A.M.D.; visualization, K.T.; supervision, G.Y.; project administration, K.T.; funding acquisition, G.Y. All authors have read and agreed to the published version of the manuscript.

### Declaration of competing interests

The authors declare no conflicts of interest. The funders had no role in the design of the study; in the collection, analyses, or interpretation of data; in the writing of the manuscript. The funders supported publishing the results.



**Acknowledgements**

We thank the Atmospheric Impacts Committee of the Coordinating Research Council (CRC) and the Electric Power
Research Institute (EPRI) for supporting this work. The findings, opinions, and conclusions are the work of the authors and
do not necessarily represent the findings, opinions, or conclusions of the CRC or EPRI. We also thank the Texas
Commission on Environmental Quality (TCEQ) for providing Harris County emissions data.

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
