# Peer review of "Technical note: Comparing ozone production efficiency (OPE) of chemical mechanisms using chemical process analysis (CPA)"

_EGUsphere, 2025_

## Author Comment (AC1)

**Technical note: Comparing ozone production efficiency (OPE) of chemical mechanisms using chemical process analysis (CPA)**

Katie Tuite, Alan M. Dunker, and Greg Yarwood

EGUSPHERE-2025-3695

We thank the referees for providing beneficial feedback, which has been addressed in a revised manuscript. Responses to individual comments from both referees are provided below.

Anonymous Referee #1

The manuscript presents a useful review of OPE from several field studies in comparison to model OPE results using several widely used photochemical mechanisms as part of a diagnostic analysis of modeled ozone concentration. EPA guidance for air quality model evaluations emphasizes that model performance evaluations (MPE) that are based on modeled and observed species concentrations (state variables) are inadequate, and that other types of MPE, including diagnostic evaluation, should be used in addition to the state variable MPE. The diagnostic analysis presented in the manuscript is useful for gaining insight into feedback processes in photochemical mechanisms and air quality models. Diagnostic model evaluations are rarely performed in regulatory applications of air quality models, and this manuscript illustrates how the chemical process analysis (CPA) in CAMx can be used for diagnostic evaluations.

Additionally, as the authors point out, most O3 nonattainment areas in the U.S. have become primarily NOx-limited for peak O3 concentrations, and thus there is a need for more model comparison metrics suitable for NOx-limited conditions. Therefore, I recommend the manuscript be accepted for publication. While I think the manuscript could be published without any major changes, I am providing several detailed comments below for the authors to consider for possible additions or clarifications to the analysis in the manuscript. I also suggest that the authors modify Table 4 to include the percent change in daily net O3 production, as described in more detail in the comment below.

**Response:** Thank you for the review and comments. We appreciate the feedback and have provided responses to specific comments below.

Line 407: "The increase in OPE-CPA at low NOx counterintuitively occurs even as O3 production decreases, and the relative changes are notably different, e.g., the OPE-CPA percent increase is 2 times larger than the O3 percent decrease at 50% ANOx. This highlights the difficulty of using OPE to predict O3 response to NOx."

Some clarification of this statement would be useful. I don't find this result to be counterintuitive because we expect that O3 production will decrease as NOx emissions are reduced (in NOx-limited regimes). We also expect that the O3 production percent decrease will be smaller than the NOx emissions percent decrease because OPE-CPA will be greater at the higher VOC/NOx ratios associated with the NOx emissions decrease.

**Response:** This statement was intended to highlight the opposing trends of OPE-CPA and $O_3$ concentration to decreased $NO_X$ emissions, and the difference in their relative responses. We have updated the statement in the manuscript to clarify.

Also, given that background O3 is a large contributor to the modeled O3 concentration, we expect that the O3 concentration percent decrease to be smaller than the O3 production percent decrease. (Although the authors constrain the model such that lateral boundary conditions do not contribute to

background O3, the residual O3 in layer two and carry over of O3 in layer one effectively acts as a source of high background O3 for each day of the simulation). In diagnostic model evaluations, it is important to distinguish between changes in O3 concentration versus changes in O3 production. If you modeled the relationship of OPE-CPE as a function of VOC/NOx, I expect that OPE-CPA would be more useful in predicting the O3 production decrease for a given NOx emissions decrease, although other feedback processes in the chemical mechanism and in the model will still make OPE-CPA an imperfect predictor of the O3 production response to NOx emissions changes.

**Response:** We agree that the contribution from background $O_3$ causes the $O_3$ concentration decrease to be smaller than the $O_3$ production decrease and have added a statement to Section 3.3 to highlight this.

Figures 5 and 6 show the relationship of OPE-CPE as a function of VOC/$NO_X$ ratio. In Figure 5, VOC is held constant and as $NO_X$ is decreased (increasing the VOC/$NO_X$ ratio), OPE-CPA increases. $O_3$ production and concentration decreases as the VOC/$NO_X$ ratio increases however (see Figure 4 and Table 4). The behaviors are anti-correlated and there is not a linear relationship between OPE-CPA and $O_3$ production which prevents OPE-CPA from being a simple predictor of $O_3$ production. The discussion about OPE-CPA and $O_3$'s relationship has been updated for clarity in Section 4.1.

Line 298: O3 response surface plots. I noted that the plots are inverted from traditional O3 isopleth plots in which the NOx scale factor is shown on the y-axis and the x-scale factor is shown on the x-axis. It is interesting that the response surface plots do not show significant reductions in ozone associated with NOx saturation even at the highest modeled NOx concentrations. This is consistent with the O3 time-series plots in Figure 2 that show morning minimum O3 concentrations around 65 ppb.  The formulation of the model is such that it never shows strong NOx titration effects with O3 concentrations approaching zero that are often seen in urban core areas with large early morning rush hour NOx emissions.  I don't think this is necessarily a problem for the analysis because O3 and NOz production are primarily controlled by the increasing mass in the PBL as its height rapidly increases in mid-morning.  However, it is worth noting that the modeled O3 time-series and O3 response plots differ from what we typically see in urban surface layer observed O3 data. It would be interesting to see OPE-CPA results for an actual 3D photochemical grid model simulation for Texas to see if the results are similar to this constrained model analysis. The results might be similar in modeled grid cells that are NOx limited, but might differ in areas that are NOx-saturated or transitional from NOx-saturated to NOx-limited.

**Response:** We agree that the model does not show strong titration effects, which may be due to how emissions are averaged in the model. Figure 1 shows the area over which model inputs are averaged, which include the entire county in which the urban center is located. Model emissions are therefore not strictly representative of the urban core. We added a statement in Section 3.2 of the manuscript to clarify this and point out the differences between the model and typical urban measured diurnal profiles.

Line 275:   Table 4 compares the effect of a 50% NOx emissions reduction on OPE-CPA and peak O3 concentration (averaged over four days).   It would also be useful to show the change in daily O3 production which is the metric more relevant to OPE-CPA.  The daily peak O3 concentration is strongly influenced by the background O3 (carry-over from the previous day as noted above).  From Figure 2, it appears that the daily morning background ozone averages about 63 ppb for all four mechanisms, and the daily net O3 Production varies from about 24 to 28 ppb. I suggest that the authors also calculate the daily net O3 production for the 50% NOx emissions reduction simulation, and include the percent change in daily net O3 production in Table 4.

**Response:** Thank you for this suggestion. We have updated Table 4 to include daily net $O_3$ production, calculated as maximum $O_3$ minus minimum $O_3$, and agree that this is a useful comparison

metric. Daily net $O_3$ production ranges from 31.0 to 35.1 ppb for the base runs and 19.2 to 22.2 ppb for the 50% $NO_X$ emissions reduction runs. The discussion of Table 4 has been updated to include daily net $O_3$ production.

Line 384: "It is unclear why a linear relationship of O3 and NOz is observed in measurements despite a nonlinear relationship between Pn(O3) and Pn(NOz) (Kleinman et al., 2002)"

I can speculate that the measurements are sampling from air parcels with similar VOC/NOx ratios, with O3 and NOz concentration being influenced primarily by dilution of the plume, so the measurement derived OPE reflects a much more compressed range of VOC/NOx ratios as compared to the scale from zero to a factor of 5 changes in the modeled NOx emissions used to generate OPE_CPA in Figure 5. However, as the authors note, more investigation is needed to understand this relationship.

**Response:** Thank you for this explanation and we agree that the measurements may not capture the same VOC/$NO_X$ ratio range that we model in this study. We hope that this work will encourage future studies to focus more specifically on how VOC/$NO_X$ variation and plume dilution impact OPE-plot.

Anonymous Referee #2

This is an interesting study that compares the performances of four widely used chemical mechanisms (CB6r5, CB7r1, SAPRC07, and RACM2) on the simulations of ozone formation by the chemical process analysis of CAMx 2-box model. Ozone Production Efficiency (OPE) was selected as a comparison metric for the NOx-limited conditions. The authors found a general consistence with considerable differences among the four mechanisms. The differences were mostly attributed to the different treatments of OH+NO2 reaction rate and nitrogen recycling chemistry. The results are useful for evaluating the existing chemical mechanisms which are critical component of atmospheric models. The manuscript is generally well written. I recommend that it can be considered for publication after the following comments being addressed.

**Response:** Thank you for the review and comments. We appreciate the feedback and have provided responses to specific comments below.

Specific comments:

Considering the significant impact of the difference in the treatment of OH+NO2 rate constant in the chemical mechanisms, the authors are strongly encouraged to provide a comment on which one is better or most appropriate for the real atmospheric conditions. This would be very helpful for the further development of chemistry mechanisms.

**Response:** Many mechanisms rely on rate recommendations from either the NASA JPL or IUPAC evaluations, but as noted in the manuscript, there is a 6% difference between two for the OH+$NO_2$=HNO3 reaction at 298 K and 1 atm. This discrepancy makes it difficult to recommend one over the other and we expect that additional rate constant measurements will be needed to narrow the difference. The availability of two comprehensive rate constant evaluations is valuable for identifying where important differences exist. We express our viewpoint with the following statement in the conclusions section: "Different rate constant recommendations from IUPAC and NASA can contribute to overall mechanism uncertainty, particularly via the important OH + $NO_2$ reaction, demonstrating that new rate constant measurements are valuable (e.g., Rolletter et al., 2025; Amedro et al., 2020) together with updated rate constant recommendations."

Lines 284-292: It is very interesting to see the different NOy budget simulated by the different mechanisms, especially the results for organic nitrogen compounds. Could the authors compare the

detailed chemical mechanisms adopted for the nitrogen compounds in these mechanisms and explain the reasons for such differences?

**Response:** Nitric acid dominates the $NO_Y$ budget (Figure 5) which moderates the influence of differences for ONs and PANs discussed here.

The differences in ON concentrations are largely impacted by how the mechanisms handle ON recycling back to $NO_X$. This is discussed in the manuscript in Sections 3.2 and 3.3 and summarized again here. RACM2 assumes all ONs recycle through reaction with OH or photolysis (see reactions 31 and 123 in Table S13), whereas the other mechanisms assume some ONs (XN in SAPRC07 and NTR2 in CB6r5/CB7r1) do not recycle. The more efficient recycling, along with higher OH concentrations, cause RACM2 to have the lowest ON concentrations. A more detailed comparison between mechanisms is complicated due to different assumptions used in the lumping schemes, including different number of ON species in each mechanism.

Model concentration of peroxyacyl nitrates (PANs) are dominated by peroxyacetyl nitrate (PAN) and higher order peroxyacyl nitrates (represented by PPN in RACM2, PAN2 in SAPRC07, and PANX in CB6r5/CB7r1). SAPRC07 predicts larger daytime concentrations of total PANs than the other mechanisms due to large concentrations of the precursor radicals (MCO3 and RCO3) needed for PANs formation. Larger MCO3 and RCO3 concentrations are due primarily to differences in VOC oxidation schemes between the mechanisms.

Section 3.2 in the manuscript was updated to include some of the details provided here.

Lines 316-318: generally, the OPE should peak or plateau at high VOC/NOx ratios (NOx-limited regime), but why this behavior is not consistent from day to day or among mechanisms?

**Response:** The concentrations vary in response to meteorology from day to day and between mechanisms so the VOC/$NO_X$ ratio is not the same in the various scenarios. The mechanisms all show a similar trend but the point at which each one peaks or plateaus is slightly different, which is due to differences in the chemistry. This statement in the manuscript has been updated for clarification.

Figure 6: I found the OPE at the VOC Factor of 0 were still larger than 6. This is unusual as the OPE values under the VOC-limited conditions (should be the case of VOC Factor = 0) are commonly much lower (e.g. 1-3). What's the reason for this?

**Response:** Figure 4 shows $O_3$ response surface plots and gives an indication of whether $O_3$ chemistry is in a $NO_X$-limited or VOC-limited regime at different emission scaling factors. At base $NO_X$ emissions (scale factor = 1) and the lowest modeled VOC emissions (scale factor = 0.1), $O_3$ chemistry is in the transition region between $NO_X$-limited and VOC-limited since $O_3$ concentration responds similarly to changes in $NO_X$ and VOC scale factor. This is consistent for all mechanisms since they each have a similar $O_3$ response plot shape, and in particular, show a similar location for the ridgeline which separates $NO_X$-limited and VOC-limited conditions. In our scaled emission runs, only anthropogenic emissions were modified, so the combination of base biogenic VOC emissions plus anthropogenic emissions scaled by 0.1 still provide enough total VOC to prevent $O_3$ chemistry from being VOC-limited.

A statement has been added to the manuscript in Section 3.3 to point out that $O_3$ is in the transition region for this emissions scenario.

Lines 338-340: again, could you comment on which one is better for the ON recycling?

**Response:** ON fate depends on heterogeneous chemistry (i.e., ON hydrolysis on/in particles) and deposition processes in addition to gas-phase reactions. Gas-phase mechanisms that resolve ON speciation in more detail provide greater opportunity for atmospheric models to resolve the influences

of heterogeneous chemistry and deposition on ON lifetime and fate. Among these mechanisms, RACM2 resolves ONs the least and CB6r5/CB7r2 the most. The $NO_X$ recycling discussion in Section 3.3 has been updated to include more detail about the effect of ON speciation.